# Stagnant zone segmentation with U-*net*

Selam Waktola*
Institute of Applied Computer Science
Lodz University of Technology
Lodz, Poland
selam.waktola@gmail.com

Krzysztof Grudzien
Institute of Applied Computer Science
Lodz University of Technology
Lodz, Poland

Laurent Babout
Institute of Applied Computer Science
Lodz University of Technology
Lodz, Poland

*Abstract*— **Silo discharging and monitoring the process for industrial or research application depend on computerized segmentation of different parts of images such as stagnant and flowing zones which is the toughest task. X-ray Computed Tomography (CT) is one of a powerful non-destructive technique for cross-sectional images of a 3D object based on X-ray absorption. CT is the most proficient for investigating different granular flow phenomena and segmentation of the stagnant zone as compared to other imaging techniques. In any case, manual segmentation is tiresome and erroneous for further investigations. Hence, automatic and precise strategies are required. In the present work, a U-net architecture is used for segmenting the stagnant zone during silo discharging process. This proposed image segmentation method provides fast and effective outcomes by exploiting a convolutional neural networks technique with an accuracy of 97 percent.**

*Keywords—U-net segmentation; deep neural networks; stagnant zone; X-ray tomography*

## I. INTRODUCTION

In many industries, like in mining, agriculture, civil engineering, and pharmaceutical manufacturing; silo is used for protecting, storing and loading granular materials into process machinery [1]. During the silo discharging process, there are two major types of flow: namely, "funnel" and "mass" flows [2]. In the case of mass flow, all granulates discharge with a uniform downward velocity across the entire cross-section area. Whereas, in the case of funnel flow, in which this paper focuses on and characterized by granular is flowing only in the center of the silo and creating a stagnant zone at the walls of the container [2,3].

The hopper geometry, internal friction between particles and wall all have a direct impact on the flow type [4]. In order to understand and describe the flow behavior and evaluating the silo wall pressures, the knowledge of the density distribution within the bulk solid is very important aspect [5]. Furthermore, during funnel flow, the shape and size of the stagnant zone depend on different factors including the granular material, the bin wall roughness, the initial packing density and filling level [6].

X-ray Computed tomography (CT) is one of the most powerful 3D imaging techniques among available tomographic techniques due to its high-resolution capability. For the last two decades CT has been used as a non-destructive method to characterize objects, visualize flows, analyze concentration changes of the bulk solid during silo discharging process [5 ,7,8]. Particle Image Velocimetry (PIV) [9,10] and Electrical Capacitance Tomography (ECT) [11–14] are also some of the techniques used to visualize concentration changes during flows of granular materials.

Even though convolutional neural network (CNN) has recently become popular and has increasingly been used as an alternative to many traditional pattern recognition problems, its application for segmenting stagnant zones of X-ray CT images is not common. For processing and analyzing process tomography data, artificial neural network algorithms were applied in electrical impedance tomography images [15–17]. This paper proposed a deep neural networks technique for segmenting the stagnant zone automatically. The main advantage of the proposed approach is an effective segmentation for acquiring the desired characteristics of flow parameters without prior image processing or expert guidance.

## II. EXPERIMENTAL SET-UP USING X-RAY TOMOGRAPHY

An especially designed model silo, with rectangular bin, allowed carrying out in situ experiments. The bin part is 10 cm wide, 5 cm deep and 20 cm high. The left and right hopper angles can be independently set to generate different types of flows, i.e. mass and funnel flows, with concentric/eccentric discharging modes. The silo model design was easily customizable to shift the outlet position and the angle between the hopper and the silo with respect to the vertical. The silo material is polycarbonate with 5 mm thickness. The outlet width is manually set, which allows controlling the discharge velocity. During concentric flow, both hoppers' angles are set to the same value. In order to observe the eccentric flow, these angles are set to a different value. The non-symmetric silo construction causes shift silo outlet to the left or right side, in the direction of the larger angle.

The time-lapse studies were performed at INSA-Lyon (France) using a GE Phoenix v|tome|x device (see Fig. 1). The device is equipped with a high energy X-ray microfocus source (up to 160 kV) with a 4 μm spot size. The detector is equipped with a 1500X1920 pixels (with a pixel size of 127X127 μm$^2$). Fig. 2 presented a picture of the silo model inside X-ray tomography hutch. During measurements, the distance between the X-ray source and the detector was equal to 577 mm and voxel size 160 um. Sorghum and rice have been used as a granular material during the experimental campaign of this study.

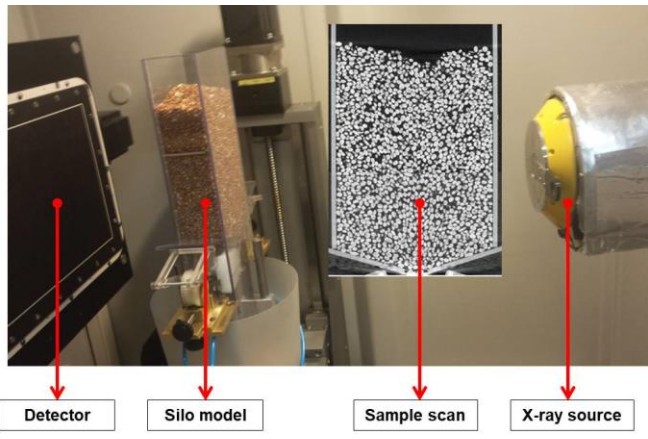

Fig. 1. X-ray Computed Tomography system with silo model.

## III. THREE-DIMENSIONAL SEGMENTATION

Cross-sectional views of the X-ray tomography of initially loosely packed rice with smooth silo bin walls are presented in Fig. 2.

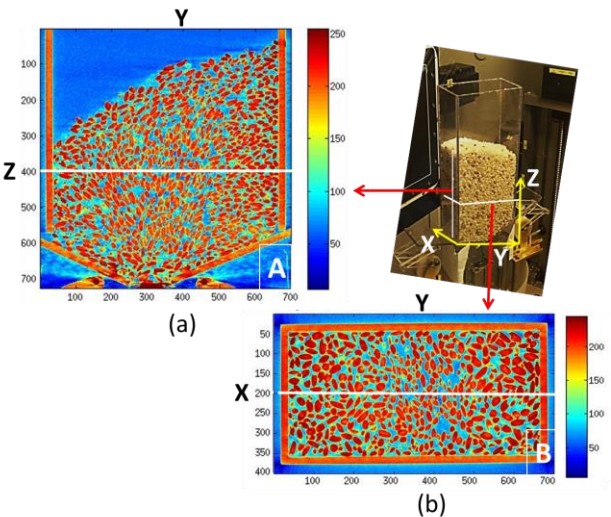

Fig. 2. (a) front side scan at x=200 voxel. (b) top side scan at z=400 voxel.

The main aim of this study was to find an effective way of segmenting the stagnant zone during eccentric discharging modes. After the initial packing density scan was performed, the outlet of the silo was opened for about 2 second and the next scan was carried out till all granulate are discharged out from the silo. Thus, to analyze the concentration changes, the absolute difference between two successive scans has been computed and it shows the formation of the stagnant zone in the case of funnel flow. Fig. 3c presents concentration changes during eccentric discharging mode (angle between the hopper and the silo with respect to the vertical axis was 30° and 20°) and it reveals the evolution of the stagnant zone at the right side of the image with a good contrast in which the hopper angle was 20°.

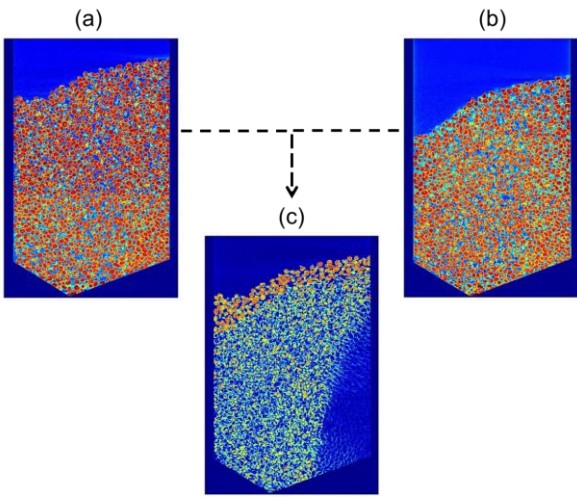

Fig. 3. Silo hopper angle: 30° on the left and 20° on the right side view in the middle of the silo where x=200 pixel. (a) Second scan. (b) Third scan. (c) The absolute difference between scan three and four of sorghum.

The last step consists in segmenting the stagnant zones. By using a three-dimensional Otsu threshold method it was possible to compute the stagnant zone mask as presented in Fig.4.

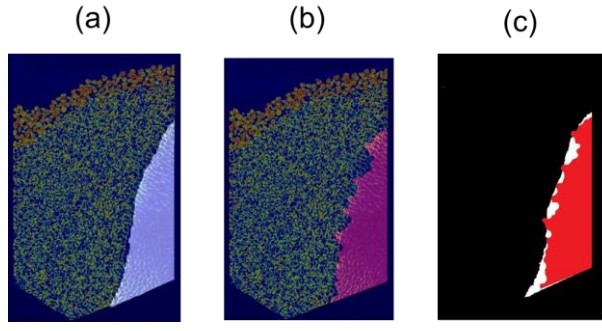

Fig. 4. (a) Ground truth at x=240 (b) segmentation using Otsu at x=240 (c) superimposing ground truth and Otsu segmentation at x=240 with red color and ground truth with white.

Although the Otsu method was able to segment the stagnant zone, it is hard to get 100% accurate segmentation out of it. Since some particles in the flow zone may shift in the position of another particle, giving an impression of no movement based on the absolute difference. Thus, such kind of effect could hinder the segmentation result. As a result, a deep neural networks approach has been applied for effective segmentation and explained hereafter.

## IV. U-NET BASED DEEP CONVOLUTIONAL NETWORKS

The main advantage of the proposed approach is an effective segmentation for acquiring the desired characteristics

of flow parameters without prior image processing or expert guidance. From several deep neural networks, u-net is one of the successful architecture which is used in different image segmentation tasks. Originally u-net neural network architecture was built for performing semantic segmentation on a small bio-medical data-set [18]. The architecture is computationally efficient and trainable with a small dataset, which is a core advantage for datasets like in material science where the little amount of labelled data is available. Despite it might be good and well-fitted in bio-medical tasks, it needs to be tuned again for fitting the granular material X-ray CT image segmentation task investigated in this study. Thus, the u-net neural network architecture and hyper-parameters values are adjusted in order to get good segmentation results.

The network has been trained on 2D segmentation and then it can find the whole 3D volume segmentation of a new scan in which the network has never seen before. As each of the CT images already contain repetitive structures with the corresponding variation, only very few images are required to train a network that generalizes reasonably well. One of the major modifications is that the original u-net used stochastic gradient descent optimizer [18], but this modified u-net architecture used Adam optimizer [19] to minimize the categorical cross-entropy objective.

During training, 40 images were selected for the training and test dataset. The dataset was first divided into two subsets, train, and test. The first subset contains 30 images in which 80% were used for training and 20% for validation. The trained model was next tested on the second subset which contains the remaining 10 images. During training, 30 manually annotated ground truth segmentations were used to train the network to recognize the stagnant zone borders. Since the available dataset is small, an extensive amount of data augmentations has been applied to improve the performance of the network. The testing datasets were used for the evaluation of the network performance.

Fig. 5 illustrates the comparison between the ground truth, segmentation by using Otsu and predicted segmentation by the modified u-net trained model for the same 2D image. As the result shows, Otsu segmentation is too sensitive for gradient changes and the trained model predicted a smooth segmentation like the ground truth.

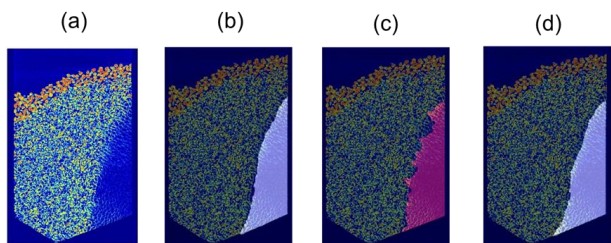

Fig. 5. (a) Original image (b) Ground truth (c) segmented by using Otsu (d) modified u-net model prediction.

Intersection over Union (IoU) is used as an accuracy measure to compare dropped out ground truth slices to the predicted results. The IoU score is a standard performance measure for the object category segmentation problem. Given a set of images, the IoU measure gives the similarity between the predicted region and the ground-truth region for an object present in the set of images and is defined as true positives/(true positives + false negatives + false positives). Table 1 presents the result of the experiment for segmenting stagnant zone.

| Network architecture | IoU |
| --- | --- |
| Original U-Net | 0.95 |
| Modified U-Net | 0.97 |

Table 1. Model performances on segmenting stagnant zone.

Once after having trained model using the CNN method, the end-to-end 3D automatic segmentation offers an effective and fast segmentation of stagnant zone during silo discharging process. The key advantage of this method could be used for deep investigation of flow characteristics without utilizing any prior image processing methods or expert guidance. For both the Otsu and the trained model execution time (on CPU) were compared for generating one segmented image. The Otsu segmentation took around 4 sec per one 2D image. Where else the trained model took less than a second for generating its prediction for a given input image.

In order to prove that the trained model (which used the sorghum grains flow as a training dataset) could generate the stagnant zone segmentation for completely different scan and grain material, it was tested by pre-processing two 3D scans of rice grains flow. The result shows that the trained model was able to generate predicted segmentation successfully. Fig. 6 presents two major steps for generating the stagnant zone prediction of new scans having similar flow property (in this case rice grains flow). The first step was computing the absolute difference of two successive scans and then this new scan was given as an input to the trained model. Finally, the trained model generates the stagnant zone segmentation as shown in Fig. 6d. Fig. 7 illustrates 3D dense segmentation of rice grains flow by superimposing the predicted mask into the original scan.

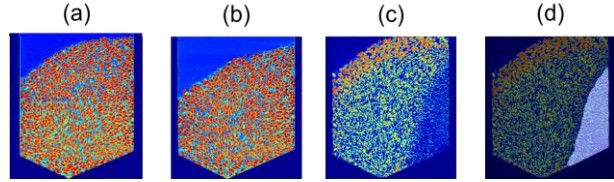

Fig. 6. (a) Rice scan 3 (b) Rice scan 4 (c) the absolute difference between scan 3 and scan 4 (d) stagnant zone prediction.

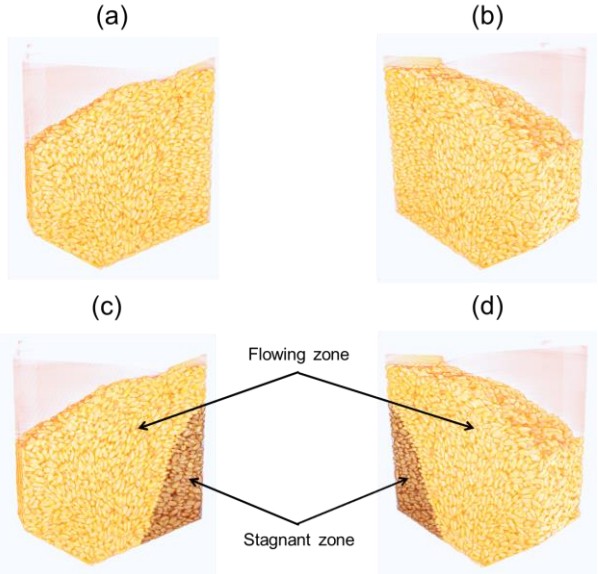

Fig. 7. Original rice grains flow scan 4 (a) front side view (b) left side view; the network predicted dense segmentation (c) ) front side view (d) left side view .

## V. CONCLUSION

In order to analyse the X-ray CT data that has been acquired from different experimental campaigns and complex structure of granular material flows, it can be tedious and extremely time-consuming for manual analysis. This paper presents a new approach for automatic segmentation of the stagnant zone in an effective way by exploiting the CNN technique. The main advantages of the proposed approach are the speed and effective segmentation for acquiring the desired characteristic flow parameters. Once having the trained model, it was tested that the model could generate a predicted segmentation in less than a second for a given completely new granular material flow image with an accuracy of 97%. The accuracy of the CNN approach could also probably be further improved if the delineations of the ground truth were acquired from different experts and more number of dataset was used. Moreover, the architecture of the model could be modified to accommodate 3D volumes of images as an input for processing them with corresponding 3D operations.

## ACKNOWLEDGMENT

The work is funded by the National Science Centre in Poland (grant number: 2015/19/B/ST8/02773).

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
