# OpenReview forum: "Stagnant zone segmentation with U-net"
_ICLR.cc/2020/Conference — Reject_

### Official Review · AnonReviewer1 · 2019-10-08
**Official Blind Review #1**

**Rating:** 1

**Review:**

- This paper simply proposes to use UNet for the segmentation of stagnant zones in X-ray CTs. While the applicability of this model may represent an advance in the particular field of the authors, the technical contribution of this paper is far from the level expected in this conference.

- As the paper reads, the main contribution of the paper is the modified version of UNet 'proposed' by the authors, which major modification consists on replacing SGD by Adam. Nevertheless, this cannot be considered a contribution, as changing the optimizer in a deep model is a marginal change, from a methodological point of view.

- Overall, the quality of the paper is below the standards of ICLR (content, technical contribution, length).

- The submission is not anonymized (authors included their names and affiliations).

**Experience Assessment:**

I have published in this field for several years.

**Review Assessment: Checking Correctness Of Derivations And Theory:**

I carefully checked the derivations and theory.

**Review Assessment: Checking Correctness Of Experiments:**

I carefully checked the experiments.

**Review Assessment: Thoroughness In Paper Reading:**

I read the paper thoroughly.

---

### Official Review · AnonReviewer2 · 2019-10-21
**Official Blind Review #2**

**Rating:** 1

**Review:**

I support desk rejection since violating double blind rule, wrong format and insufficient length.





**Experience Assessment:**

I have published one or two papers in this area.

**Review Assessment: Checking Correctness Of Derivations And Theory:**

N/A

**Review Assessment: Checking Correctness Of Experiments:**

N/A

**Review Assessment: Thoroughness In Paper Reading:**

I made a quick assessment of this paper.

---

### Official Review · AnonReviewer3 · 2019-10-30
**Official Blind Review #3**

**Rating:** 1

**Review:**

This paper propose a modified U-net architecture to segment the stagnant zone during silo discharging process. It lacks novelty and the improvement is marginal.  More importantly than all of that, this paper violates the double blind review rule and is same with [1]. So I think this paper is not suitable for acception.

[1]Waktola S, Grudzien K, Babout L. Stagnant zone segmentation with U-net[C]//2019 IEEE Second International Conference on Artificial Intelligence and Knowledge Engineering (AIKE). IEEE, 2019: 277-280.


**Experience Assessment:**

I have read many papers in this area.

**Review Assessment: Checking Correctness Of Derivations And Theory:**

I carefully checked the derivations and theory.

**Review Assessment: Checking Correctness Of Experiments:**

I carefully checked the experiments.

**Review Assessment: Thoroughness In Paper Reading:**

I read the paper at least twice and used my best judgement in assessing the paper.

---

### Comment · AnonReviewer2 · 2019-10-21
**Desk rejection**

This paper contains authors name and affiliation, and violates the double blind review rule.

---

### Decision · Program_Chairs · 2019-12-19

**Decision:**

Reject

**Comment:**

The paper proposed U-net for segmentation of stagnant zones in computed tomography. Technical contribution of the paper is severely limited, and is not of the quality expected of publications in this venue. The paper is not anonymized and violates the double blind review rule. I'm thus recommending rejection.